# OpenReview forum: "Large Language Models Often Say One Thing and Do Another"
_ICLR.cc/2025/Conference — ICLR 2025 Poster_

### Official Review · Reviewer_8NCb · 2024-10-16

**Soundness:** 2
**Presentation:** 2
**Contribution:** 2
**Rating:** 3
**Confidence:** 4

**Summary:**

This paper introduces an evaluation benchmark to assess the consistency between the verbal outputs of language models and fact-based realities that goes by the Words and Deeds Consistency Test (WDCT).

This benchmark ensures the correspondence between word-based and fact-based questions in various domains. Evaluations with the GPT, Llama and Mistral-based models revealed weaknesses, potentially due to model tuning problems.

This research highlights the importance of assessing the reliability of LLMs by measuring the alignment between words and actions.

**Strengths:**

The contribution has its merits. A detailed list follows:

- The writing is clear, fluent and understandable. However, the organisation of the introduction should definitely be reconsidered (comment on weak points)

- The authors address an important issue in evaluating words and deeds using LLMs.

- The datasets introduced in the paper include several domain topics to ensure better generalisability (however, see weaknesses).

The habitual evaluation of LLMs is very extensive and detailed.

**Weaknesses:**

Although the contribution has its myths, there are many points that need to be clarified and improved:

**Introduction**

- RQs are described (this is a good thing). However, these should be dealt with in the course of the contribution and perhaps discursively introduced in the intro. The ‘answers’ delivered by the authors in the intro to lines 49,53,73 should be revised (this is not a note on the content but on the style ‘To answer...’ is not a proper formula to be repeated three times).

- The summary of contributions on line 82 makes sense but why use line 49 to line 77 to answer the RQs and then repeat lines 82-90?

**Section 2 and 3**

- The experimental setting is described in a sufficient and comprehensible manner, even if it is sometimes difficult to follow the discourse due to the positioning of the figures.

- The evaluation is made on a limited dataset (1325 examples). This is not a major problem, but it does undermine the scientific soundness of the paper. For example, look at the well-known QA tasks (SIQA, PIQA) or the mathematicians (GSM8K). all of these have significantly more instances than 1k. It would be interesting to observe these phenomena on a larger scale (e.g. 4k).

- The construction of the examples could be biased by the constructor (an LLM constructs, to be precise does augmentation, of the examples on which it is prompted)


**Section 4 and 5**

The fact that the authors decided to organise the section into Discoveries and Discovery Discussions is not wrong as many papers do this. However, I do not think it is the best way. Indeed, the contribution shows a lot of analysis but it is really limiting to organise the presentation of results in this way.

Furthermore, it is difficult to read the comments of the experiments in the findings. For example, see ‘Exp. 1’ on line 326. I understand that the authors wanted to make the work more discursive, but they should pay more attention to style. An example could be constructing subsections by structuring the discourse by placing the RQ next to the answer. The title of the section should describe the analysis (e.g. line 334 should be a subsection called ‘inconsistency’ or something similar).



I would strongly advise the authors to revise the structure of the paper as in this condition it is really a pity to leave it as it is.

**Questions:**

- What is the role of the consistency score? As a non-multiple-choices question answer, your score should be able to capture all possible generations. Did you test and or ablation test for this?

- How much impact do systemic demonstrations in SFT have on DPO?

- What are the limits of the transferability of your study? What if another LLM was used instead of GPT-4 to build the samples?

---

> ### Author Response · Authors · 2024-11-21
> **Rebuttal (part 1/3)**
>
> Dear Reviewer,
>
> Thanks for your detailed and constructive comments, which have been really valuable in enhancing the quality of the paper. Below you will find our point-by-point response to your comments or suggestions. We hope the revisions and explanations will be satisfactory and address your previous concerns.
>
> > **C1:** RQs are described (this is a good thing). However, these should be dealt with in the course of the contribution and perhaps discursively introduced in the intro. The ‘answers’ delivered by the authors in the intro to lines 49,53,73 should be revised (this is not a note on the content but on the style ‘To answer...’ is not a proper formula to be repeated three times).
>
> Thanks for your valuable suggestion.
>
> In the revised manuscript, we gradually introduce the research questions by discussing the prevailing background and existing research gaps, naturally leading to the issue of LLMs’ consistency between words and deeds **(line35-42 in new PDF)**. This makes the logical progression of the questions much clearer, moving from the current situation (Are LLMs consistent in words and deeds?) to the cause (If not, what role does alignment play in this context?) and lastly to the method (can common knowledge generalization methods facilitate consistency between LLMs' words and deeds?).
>
> For the answers to the research questions, we also abide by the progressive relationship between the research questions, allowing for a more natural transition **(line 51-83 in new PDF)**.
>
> > **C2:** The summary of contributions on line 82 makes sense but why use line 49 to line 77 to answer the RQs and then repeat lines 82-90?
>
> We have reorganized our contributions, focusing on succinctly describing the practical and theoretical significance of the paper, such as proposing a novel word-deed consistency evaluation benchmark, instead of reiterating our answers to the research questions **(line 87-97 in new PDF)**.
>
> > **C3:** The experimental setting is described in a sufficient and comprehensible manner, even if it is sometimes difficult to follow the discourse due to the positioning of the figures.
>
> We apologize for any unpleasant reading experience you may have had.
>
> However, the figures and tables in the experimental setting section all serve to present the details of the experimental setting of the paper, such as the specific prompt, results of grid search of learning rates. Considering that these details are not the core issues of our research and there is a strict space limit for the paper, we had no choice but to put them in the appendix.
>
> In order to offer a clearer reading experience, we have revised the original text, placing them at the end of the paragraph and emphasizing that the figures and tables present experimental details, placed in the appendix **(as shown in section 3.3)**.
>
> > **C4:** The fact that the authors decided to organise the section into Discoveries and Discovery Discussions is not wrong as many papers do this. However, I do not think it is the best way. Indeed, the contribution shows a lot of analysis but it is really limiting to organise the presentation of results in this way.
>
> We respect your opinion, however, we believe the structuring into ‘Discoveries’ and ‘Discovery Discussions’ is justified as **the two sections discuss two distinct analyses**. Specifically, the ‘Discoveries’ (section 4) is essentially the answers to research questions. We have followed the logical progression of these questions and elucidated the experimental conclusions separately. The ‘Discovery Discussions’ (section 5) is our robustness analysis of the results. Here, we have conducted a series of critical analyses to eliminate the influence of factors unrelated to word and deed differences, primarily to ensure the reliability of our results. Therefore, we strongly felt that it was necessary to discuss these in two separate sections.

---

> ### Author Response · Authors · 2024-11-21
> **Rebuttal (part 2/3)**
>
> > **C5:** Furthermore, it is difficult to read the comments of the experiments in the findings. For example, see ‘Exp. 1’ on line 326. I understand that the authors wanted to make the work more discursive, but they should pay more attention to style. An example could be constructing subsections by structuring the discourse by placing the RQ next to the answer. The title of the section should describe the analysis (e.g. line 334 should be a subsection called ‘inconsistency’ or something similar).
>
> Thanks for your constructive suggestion. Based on your suggestion, we have undertaken a logical restructuring of Section 4. This section is organized into three subsections, each corresponding to one of the research questions **(section 4.1, 4.2, 4.3 in new PDF)**. Each subsection now contains multiple sub-conclusions and one overall conclusion. Each sub-conclusion is presented as a paragraph, with the title of the paragraph describing the analysis (for example, "Inconsistency between LLMs' words and deeds"). The first sentence of each paragraph states sub-conclusion (for example, “Inconsistency between words and deeds is a common phenomenon across LLMs and domains”), followed by a detailed description and analysis of the experimental results. The overall conclusion **(at the end of section 4.1, 4.2, 4.3 in new PDF)** summarises the aforementioned sub-conclusions and offers answer to the corresponding research question. We believe this new structure will facilitate better reading and comprehension of our work.
>
> > **C6:** The evaluation is made on a limited dataset (1325 examples). This is not a major problem, but it does undermine the scientific soundness of the paper. For example, look at the well-known QA tasks (SIQA, PIQA) or the mathematicians (GSM8K). all of these have significantly more instances than 1k. It would be interesting to observe these phenomena on a larger scale (e.g. 4k).
>
> Thanks for the suggestion.
>
> However, the construction of the dataset is quite labor-intensive. Each test case pair is created based on different beliefs which require manual selection from various datasets (as shown in section 2.3.1). Additionally, a certain proportion of these crafted test cases require rewriting and all test cases need to be double-checked (as shown in section 2.3.4). This makes it challenging to rapidly expand the size of our dataset in a short period of time.
>
> Despite this, we provided an analysis of the correlation between the data size and the evaluation results **in the last discussion of section 5**.  Specifically, we randomly sampled the dataset five times at various sample ratios and compared the results on the subsets with the results on the complete test set. The consistency scores on subsets of the test set at different sample ratios are listed **in Appendix B.1 in the new PDF**. We can observe that: 1) models’ consistency scores largely stabilize when the data size exceeds 50% ( as shown in Figure 10). 2) there are no statistically significant differences ($p > 0.05$) between the evaluations performed on the subsets and the entire dataset (as shown in Table 8).
>
> Therefore, we reasonably infer that evaluations based on 1,000+ test cases are stable and consistent and the phenomena observed would remain consistent on a larger scale.

---

> ### Author Response · Authors · 2024-11-21
> **Rebuttal (part 3/3)**
>
> > **C7:** The construction of the examples could be biased by the constructor (an LLM constructs, to be precise does augmentation, of the examples on which it is prompted)
>
> We understand your concern about the potential bias from the LLM constructor. However, in our study, we primarily rely on the exceptional language generation abilities of the LLM to transform manually constructed word questions into deed questions. Moreover, the test cases constructed by the LLM are just the initial versions, which are then manually reviewed and rewritten where necessary. This is a common practice in benchmark construction, such as SOTOPIA``[1]``, an evaluation benchmark for social intelligence that prompts GPT-4 to generate characters, relationships, and scenarios in test cases; and Mercury``[2]``, a code efficiency benchmark that constructs test case generators using GPT-4.
>
> To alleviate your concern, we also constructed test cases using qwen2.5-72b-instruct following the same procedure presented in the paper. The results show a persistent and significant inconsistency between words and deeds across LLMs and domains, demonstrating that **our findings are not dependent on the constructor**. Moreover, the Pearson correlation coefficient between the scores obtained from the original and new test cases is 0.95 > 0.8, further verifying the robustness of our results.
>
> | Model                  | consistency score on test cases generated by GPT-4 | by Qwen |
> |------------------------|-------|------|
> | GPT-4                  | 0.81  | 0.79 |
> | GPT-3.5-Turbo          | 0.72  | 0.72 |
> | Vicuna-7B              | 0.57  | 0.56 |
> | Vicuna-13B             | 0.57  | 0.58 |
> | Vicuna-33B             | 0.64  | 0.68 |
> | Llama-2-7B             | 0.44  | 0.52 |
> | Llama-2-13B            | 0.53  | 0.54 |
> | Llama-2-7B-Chat        | 0.54  | 0.55 |
> | Llama-2-13B-Chat       | 0.53  | 0.60 |
> | Mistral-7B             | 0.61  | 0.65 |
> | Mistral-7B-Instruct    | 0.63  | 0.68 |
> | Chatglm3-6B-Base       | 0.68  | 0.70 |
> | Chatglm3-6B            | 0.52  | 0.53 |
>
> > **Q1:** What is the role of the consistency score? As a non-multiple-choices question answer, your score should be able to capture all possible generations. Did you test and or ablation test for this?
>
> Thanks for your question. We’d like to clarify that all our test cases are multiple-choice questions (as shown in the examples in Table 1). Therefore, we utilize regular matching to extract options from the model's free-text responses. The consistency score refers to the proportion of test cases where LLMs’ choices to the word question and deed question is consistent (as described in section 3.2.2).
>
> > **Q2:** How much impact do systemic demonstrations in SFT have on DPO?
>
> Thank you for your question. I'm a bit confused by "systemic demonstrations in SFT". In terms of the SFT process, we utilize standard system messages and 0-shot questions for training the model. We believe this process is set realistically and should not affect the experimental conclusions regarding DPO.
>
> > **Q3:** What are the limits of the transferability of your study? What if another LLM was used instead of GPT-4 to build the samples?
>
> Thanks your thoughtful question. We believe our work has great transferability, considering the consistency of the results based on multiple LLM constructors as shown **in reply to C7**, and various robustness analyses **in Section 5**. If a different LLM, other than GPT-4 was used to build the samples, it's likely that similar results would be obtained.
>
> **Reference**
>
> ``[1]`` SOTOPIA: Interactive Evaluation for Social Intelligence in Language Agents. ICLR, 2024.
>
> ``[2]`` Mercury: A Code Efficiency Benchmark for Code Large Language Models . NeurIPS, 2024.

---

> ### Author Response · Authors · 2024-11-25
> **A kind reminder**
>
> Dear Reviewer 8NCb,
>
> With the discussion phase ending **in 2 days**, we sincerely hope to receive your response. If you have any questions or concerns regarding our explanation, please do not hesitate to contact us. Your response is of great significance to us for improving this work, and we look forward to hearing from you.
>
> Warm Regards, :-)

---

> > ### Comment · Reviewer_8NCb · 2024-11-25
> > **Answer**
> >
> > Thank you very much for your answers. The additional experiments sound good, however the issues of writing and writing remain open.
> >
> > I would also like to point out that although the experiments in question can be suond these are really many and change the identity of the current paper.

---

> > > ### Author Response · Authors · 2024-11-26
> > > **Looking forward to a more detailed discussion**
> > >
> > > Dear Reviewer 8NCb,
> > >
> > > Thank you for your feedback.
> > >
> > > 1. In accordance with ICLR's rebuttal policy, **we have revised and uploaded a new version of our paper based on your comments on the writing**. If there are any other issues with the writing that you feel we need to address, please do not hesitate to point them out to us.
> > >
> > > Note: we have made minor adjustments to section 4 today, to make the conclusions more reader-friendly.
> > >
> > > 2. **The additional experiments added do not impact the original contributions and conclusions of the paper, rather they provide more evidence**. Moreover, ICLR encourages revisions to the PDF to make the paper more robust, which we also believe is the purpose of the discussion stage.
> > >
> > > Specifically, we have added two experiments in response to your rebuttal:
> > > - Build test cases based on other LLM constructors. The experiment confirmed that our benchmark is not biased and our findings are independent of the constructor.
> > > - Analysis on the correlation between data size and evaluation results. The experiment confirmed the sufficiency of our benchmark's data size.
> > >
> > > 3. We appreciate your comments which have helped improve our paper. If there are other issues, we welcome you to point them out.
> > >
> > >
> > > Your feedback is very important to us, and we look forward to hearing from you!
> > >
> > > Best regards.

---

> > > ### Author Response · Authors · 2024-12-02
> > > **A summary of the major revisions**
> > >
> > > Dear Reviewer 8NCb,
> > >
> > > I hope this message finds you well!
> > >
> > > We have summarized our main revisions below, in hopes of saving you time and effort during the review process:
> > >
> > > **Writing**
> > > - Updated the introduction of the research question and its answer along with the contribution from different angles in section 1.
> > >
> > > - Updated the presentation of section 4, placing conclusions after research questions. The conclusions are organized into main conclusions (conclusion 1-3) and sub-conclusions (finding 1-6) for ease of comprehension.
> > >
> > > - A more comprehensive discussion on the related works in section 6.
> > >
> > > - Corrected the layout of tables and figures.
> > >
> > > **Experiments**
> > > - Results of more recent, advanced models, the LLaMA-3-8B and LLaMA-3-8B-Instruct in Table 3.
> > >
> > > - A data augmentation method by automatically generating aligned data in section 4.3.2.
> > >
> > > - An analysis of the correlation between the data size and evaluation results in Appendix B.1.
> > >
> > > - An analysis of the impact of LLM constructors.
> > >
> > > - An analysis of the sensitivity of the consistency scores to the prompts.
> > >
> > > Finally, we'd like to kindly remind you that the discussion phase will conclude **in 1 day**, and we sincerely hope to receive your response.
> > >
> > > Best Regards

---

> > > ### Author Response · Authors · 2024-12-03
> > > **Awaiting Your Confirmation**
> > >
> > > Dear Reviewer 8NCb,
> > >
> > > With the discussion phase ending soon, we would greatly appreciate any further guidance or confirmation from your side. We wanted to follow up on our previous response to your valuable suggestions. Your insights are very important to us, and we are eager to ensure that we have addressed all concerns appropriately.
> > >
> > > Warm Regards, :-)

---

### Official Review · Reviewer_vDJr · 2024-10-24

**Soundness:** 3
**Presentation:** 2
**Contribution:** 2
**Rating:** 6
**Confidence:** 4

**Summary:**

1. This work proposes a novel benchmark called the WDCT to measure the inconsistency between words and actions. This benchmark contains test pairs of word-based and action-based questions across various domains to measure the alignment between models' statements and their behaviors.

2. The paper introduces the issue of large language models (LLMs) being inconsistent between what they say (words) and what they do (deeds). This inconsistency is observed across different domains such as opinions, non-ethical values, and theories. The authors argue that this gap reduces trust in LLMs.

3. The authors introduce two metrics, the Consistency Score (CS) and the Probability Consistency Score (PCS), to evaluate the alignment between words and deeds. These metrics help quantify how well models maintain consistency across different tasks.

4. Conducted extensive experiments on multiple state-of-the-art LLMs, including GPT-4, Vicuna, and LLaMA, to evaluate their consistency using the WDCT. The results show a significant misalignment between words and deeds across models and domains, particularly in non-ethical value contexts.

I am willing to increase my score during the rebuttal phase if the authors address my concerns.

**Strengths:**

1. The paper introduces two new metrics, Consistency Score (CS) and Probability Consistency Score (PCS), which provide a quantitative way to measure how well LLMs maintain consistency between their words and deeds. These metrics offer valuable tools for evaluating model alignment.

2. Conduct thorough experiments on various LLMs, including GPT-4, Vicuna, and LLaMA, across various domains like opinions, non-ethical values, and theory. The experiments reveal that all the tested models exhibit the issue of inconsistency between what they say and what they do across all the domains.

3. This work shows that commonly used methods such as COT, alignment and paraphrasing do not solve this issue.

**Weaknesses:**

1. The paper looks a little bit rough in places. For example, Table 4 and Figure 8 are poorly formatted, extending beyond the usual layout boundaries, which affects readability and professionalism.

2. While the paper effectively identifies the issue of inconsistency between words and deeds in LLMs, it does not provide any in-depth analysis or reasoning behind why this inconsistency occurs. This leaves a gap in understanding the reason cause of the problem.

3.  Although some alignment techniques are explored, they are not shown to fully resolve the inconsistency, and the paper does not provide clear insights for overcoming this challenge.

4. The results rely on somewhat outdated models like LLaMA 2. Given that more advanced versions like LLaMA 3 and LLaMA 3.1 are now available, it would have been better to test these newer models to assess whether they exhibit similar issues or show improvements in consistency.

5. This work lacks ablation studies. Like studying which components of the models or alignment techniques contribute most to the word-action inconsistency. This would help clarify whether the problem is primarily due to data, model architecture, or training techniques.

**Questions:**

1. In Table 3, both LLaMA 2 and Mistral base models show relatively low average Consistency Scores (CS), whereas ChatGLM3 base has a noticeably higher CS. Does this suggest some fundamental differences in the architecture, training methodology, or other factors that set ChatGLM3 apart from the other two models?

2. In Figure 5, the consistency rate for the second epoch suddenly drops from over 50% in epoch 1 to 0% in epoch 2. Is there anything unusual about this drastic drop? Could it be related to the specific alignment process, or does this indicate a possible issue in the model's training or evaluation setup?

3.  Given that LLaMA 2 is somewhat outdated, would it be possible to try the newer LLaMA 3.1 Instruct model? It would be interesting to see if the consistency issues observed in LLaMA 2 are still present in LLaMA 3.1 instruct.

4. Have you analyzed how sensitive the consistency scores are to the prompts?

---

> ### Author Response · Authors · 2024-11-22
> **Rebuttal (part 1/3)**
>
> Dear Reviewer,
>
> Thank you for your insightful and constructive feedback. Below you will find our point-by-point response to your comments or suggestions. We hope our revisions and explanations will fully address your previous concerns and be satisfactory.
>
> > **C1:** The paper looks a little bit rough in places. For example, Table 4 and Figure 8 are poorly formatted, extending beyond the usual layout boundaries, which affects readability and professionalism.
>
> Thanks for your kind reminder. We have thoroughly reviewed the manuscript and made adjustments to the formatting, particularly the placement of figures and tables. The formatting of Table 4 and Figure 8 has been improved.
>
> > **C2:** While the paper effectively identifies the issue of inconsistency between words and deeds in LLMs, it does not provide any in-depth analysis or reasoning behind why this inconsistency occurs. This leaves a gap in understanding the reason cause of the problem.
>
> Thanks for your thoughtful review.
>
> But we’d like to clarify that in our paper, we provided an analysis of the reasons behind the inconsistency. Specifically, **in section 4.1, we offer an initial hypothesis for the reason**: for base models, it’s mainly due to a lack of strong belief and for aligned models, it may stem from the difficulty of generalizing one-sided alignment to the other side. Furthermore, **in section 4.2, we verified the reasons behind the inconsistencies in the aligned model's words and deeds**.
>
> We have restructured the organization of our findings in section 4, to present our experimental conclusions more clearly.
>
> > **C3:** Although some alignment techniques are explored, they are not shown to fully resolve the inconsistency, and the paper does not provide clear insights for overcoming this challenge.
>
> Thanks for your insightful comment.
>
> We’d like to acknowledge that the main goal of our study is to establish a benchmark for quantifying LLMs’ inconsistency between words and deeds, and provide some widely-used baselines to identify the universality and challenges of the problem. Solving this problem is not our primary concern in this paper.
>
> Further, we’d like to provide some insights for overcoming this challenge, based on our findings in section 4.2, that one-sided alignment is difficult to generalize to the other side.
> - From the perspective of data, a natural solution is to automatically generate aligned training data from one-sided data, and co-train the model with both sets of data. **We validated the effectiveness of autogenerated aligned data for enhancing the consistency using the data construction pipeline proposed in our paper, as detailed in section 4.3 of the new PDF.** The results are as follows.
> | Model                            | CS After aligning on repeated words | CS After aligning words and generated deeds by tested model itself | CS After aligning words and generated deeds by Qwen2.5-72B-Instruct |
> |----------------------------------|----------------------------|---------------------------------------------------------------------|--------------------------------------------------------------------|
> | Llama-2-7B-Chat                 | 0.53                       | 0.55                                                                | **0.63**                                                               |
> | Mistral-7B-Instruct             | 0.71                       | 0.77                                                                | **0.86**                                                               |
> | Chatglm3-6B                     | 0.62                       | 0.65                                                                | **0.69**                                                               |
>
> - From the perspective of model, we infer that this problem fundamentally lies in models’ low utility of knowledge. This would require more exploration, which we leave for future work.
>
> > **C4:**: The results rely on somewhat outdated models like LLaMA 2. Given that more advanced versions like LLaMA 3 and LLaMA 3.1 are now available, it would have been better to test these newer models to assess whether they exhibit similar issues or show improvements in consistency.
>
> Thanks for the suggestion. We have now included results from more recent, advanced versions of the LLaMA model. The results show that **while inconsistencies have been somewhat reduced in the latest LLaMA versions, the problem still persists**.
>
> | Model                    | CS   |
> |--------------------------|-------|
> | Llama-2-7B               | 0.44  |
> | Llama-2-13B              | 0.53  |
> | Llama-2-7B-Chat          | 0.54  |
> | Llama-2-13B-Chat         | 0.53  |
> | Llama-3-8B               | 0.68  |
> | Llama-3-8B-Instruct      | 0.65  |
> | Llama-3.1-8B             | 0.68  |
> | Llama-3.1-8B-Instruct    | 0.69  |

---

> ### Author Response · Authors · 2024-11-22
> **Rebuttal (part 2/3)**
>
> > **C5:** This work lacks ablation studies. Like studying which components of the models or alignment techniques contribute most to the word-action inconsistency. This would help clarify whether the problem is primarily due to data, model architecture, or training techniques.
>
> Due to the complexity of LLM training and considering the limited reply time, it is challenging for us to thoroughly study the impact of all factors during the training process on LLMs’ consistency of words and deeds.
>
> To simplify, we started from two base models, performed supervised fine-tuning on three datasets, and aligned models by two alignment techniques to investigate the impact of different factors*. The final word-deed consistency scores of the models are shown in the following table.
>
> **We can observe that:**
> - Data, model architecture, and training techniques all influence the consistency between words and deeds. We also noted interactions between these factors.
> - The problem of word-deed inconsistency is common across models, data, and alignment methods.
>
> Therefore, we infer that the root cause of word-action inconsistency may lie in the pre-training stage or the basic model architecture, indicating that it is a universal issue.
>
>
> | Model        | SFT Data         | Alignment Method | CS of Base Model | CS of SFT Model | CS of Aligned Model |
> |--------------|------------------|------------------|------------|-----------|---------------|
> | Llama-3-8B   | ultraChat``[1]``        | DPO``[4]``              | 0.68       | 0.68      | 0.50           |
> | Llama-3-8B   | OpenHermes-2.5``[2]``   | DPO              | 0.68       | 0.67      | 0.54          |
> | Llama-3-8B   | No Robots``[3]``        | DPO              | 0.68       | 0.68      | 0.57          |
> | Llama-3-8B   | ultraChat        | CoH``[5]``              | 0.68       | 0.68      | 0.54          |
> | Llama-3-8B   | OpenHermes-2.5   | CoH              | 0.68       | 0.67      | 0.38          |
> | Llama-3-8B   | No Robots        | CoH              | 0.68       | 0.68      | 0.49          |
> | Qwen2.5-7B   | ultraChat        | DPO              | 0.64       | 0.65      | 0.42          |
> | Qwen2.5-7B   | OpenHermes-2.5   | DPO              | 0.64       | 0.64      | 0.35          |
> | Qwen2.5-7B   | No Robots        | DPO              | 0.64       | 0.66      | 0.43          |
> | Qwen2.5-7B   | ultraChat        | CoH              | 0.64       | 0.65      | 0.51          |
> | Qwen2.5-7B   | OpenHermes-2.5   | CoH              | 0.64       | 0.64      | 0.48          |
> | Qwen2.5-7B   | No Robots        | CoH              | 0.64       | 0.66      | 0.54          |
>
>
> *In the SFT stage, we adopted a learning rate of 5e-6. In the alignment stage, we tried learning rates of [5e-6, 1e-6, 5e-7] and chosen the best performing ones.  For CoH, we include 'p,n,pn,np' four types of feedback and use default other parameters.
>
>
> **Reference:**
>
> ``[1]`` https://huggingface.co/datasets/stingning/ultrachat
>
> ``[2]`` https://huggingface.co/datasets/teknium/OpenHermes-2.5
>
> ``[3]`` https://huggingface.co/datasets/HuggingFaceH4/no_robots
>
> ``[4]`` Direct preference optimization: Your language model is secretly a reward model. NeurIPS, 2023.
>
> ``[5]`` Chain of Hindsight aligns Language Models with Feedback. ICLR, 2024.

---

> > ### Comment · Reviewer_vDJr · 2024-11-22
> > **Thanks for your reply**
> >
> > I have increase the score accordingly. Thanks.

---

> ### Author Response · Authors · 2024-11-22
> **Rebuttal (part 3/3)**
>
> > **Q1:** In Table 3, both LLaMA 2 and Mistral base models show relatively low average Consistency Scores (CS), whereas ChatGLM3 base has a noticeably higher CS. Does this suggest some fundamental differences in the architecture, training methodology, or other factors that set ChatGLM3 apart from the other two models?
>
> Thanks for your insightful question.
>
> As the training data and details of models aren't publicly available, we can only speculate based on the technical reports and experimental results of the models.
>
> **We hypothesize two possible reasons:**
> - From the experimental results in Table3, it appears that ChatGLM3 has a significant advantage in value domain. Hence, we infer that this may be due to ChatGLM3 pay more attention on value-related training in the pre-training phase, such as removing texts contrary to mainstream values during data selection. The original text from the technical report states: "cleaned data in the pre-training stage by removing text containing sensitive keywords and web pages from a pre-defined blacklist."``[1]``
> - The training objectives of ChatGLM``[2]`` differ from those of LLaMA2``[3]`` and Mistral``[4]``. GLM is trained by optimizing an autoregressive blank infilling objective, , while LLaMA and Mistral are trained on the next token prediction task. This differential in model structures could potentially contribute to the noted inconsistency.
>
> > **Q2:** In Figure 5, the consistency rate for the second epoch suddenly drops from over 50% in epoch 1 to 0% in epoch 2. Is there anything unusual about this drastic drop? Could it be related to the specific alignment process, or does this indicate a possible issue in the model's training or evaluation setup?
>
> Thanks for your valuable feedback. In Figure 5, the y-axis represents the final consistency score of questions newly aligned in this round. The sudden drop in the consistency rate for the second epoch signifies that no new questions were aligned after the second round.
>
> > **Q3:** Given that LLaMA 2 is somewhat outdated, would it be possible to try the newer LLaMA 3.1 Instruct model? It would be interesting to see if the consistency issues observed in LLaMA 2 are still present in LLaMA 3.1 instruct.
>
> Thanks for your suggestion. We’ve included the results of LLaMA 3.1 instruct, and you can refer to the results in the reply to C4.
>
> > **Q4:** Have you analyzed how sensitive the consistency scores are to the prompts?
>
> Thanks for your question.
>
> We have analyzed the sensitivity of the LLM's choices to prompts in section 5, finding that LLM's choices are relatively consistent across paraphrased prompts, a conclusion in accordance with previous work. The consistency score analysis naturally tends to be stable if the LLM's choices are stable.
>
> To further solidify our experimental results, **we supplemented an analysis of the sensitivity of the consistency scores to the prompts**. Specifically, we carried out five paraphrases of the test cases based on GPT-4, and separately evaluated models’ consistency scores. The specific scores and standard deviations are as follows, showing that the consistency scores, in general, remain relatively stable even when the prompts are changed.
>
> | Model                | Result1 | Result2 | Result3 | Result4 | Result5 | Avg CS | STD  |
> |----------------------|---------|---------|---------|---------|---------|--------|------|
> | GPT-4                | 0.80    | 0.82    | 0.82    | 0.81    | 0.85    | 0.82   | 0.02 |
> | GPT-3.5-Turbo        | 0.74    | 0.72    | 0.71    | 0.70    | 0.73    | 0.72   | 0.02 |
> | Llama-2-7B           | 0.55    | 0.54    | 0.55    | 0.56    | 0.53    | 0.55   | 0.01 |
> | Llama-2-7B-Chat      | 0.55    | 0.52    | 0.53    | 0.53    | 0.53    | 0.53   | 0.01 |
> | Mistral-7B           | 0.64    | 0.64    | 0.64    | 0.64    | 0.66    | 0.64   | 0.01 |
> | Mistral-7B-Instruct  | 0.63    | 0.63    | 0.61    | 0.63    | 0.64    | 0.63   | 0.01 |
> | Chatglm3-6B-Base     | 0.68    | 0.68    | 0.68    | 0.68    | 0.64    | 0.67   | 0.02 |
> | Chatglm3-6B          | 0.51    | 0.50    | 0.51    | 0.54    | 0.54    | 0.52   | 0.02 |
>
>
> **Reference:**
>
> ``[1]`` Chatglm: A family of large language models from glm-130b to glm-4 all tools.
>
> ``[2]`` GLM: General Language Model Pretraining with Autoregressive Blank Infilling. ACL, 2022.
>
> ``[3]`` Llama 2: Open foundation and fine-tuned chat models.
>
> ``[4]`` Mistral 7B.

---

### Official Review · Reviewer_eRDy · 2024-10-31

**Soundness:** 4
**Presentation:** 3
**Contribution:** 3
**Rating:** 8
**Confidence:** 3

**Summary:**

The paper addresses a critical issue within LLMs — the inconsistency between their expressed words (statements or opinions) and their actions (behavioral outputs). To quantify this inconsistency across various domains, the authors introduce a novel evaluation benchmark, the Words and Deeds Consistency Test (WDCT). Through the WDCT, the study reveals substantial discrepancies between what LLMs say and do, emphasizing the challenges in achieving consistent alignment, especially when alignment is performed only on words or deeds independently. The paper makes a valuable contribution by suggesting a new direction for alignment research in LLMs.

**Strengths:**

- Relevance and Importance:
The problem tackled is highly relevant. As LLMs are increasingly embedded in real-world applications where consistency between communication and behavior is essential, identifying and addressing this issue has meaningful implications for enhancing user trust and model reliability, particularly as LLMs are now frequently deployed as autonomous agents.

- Novel Evaluation Benchmark (WDCT):
The WDCT is a significant contribution, offering a well-structured approach to measure alignment between words and deeds across multiple domains (e.g., opinion, ethical values, theory). This benchmark provides a robust foundation for future alignment studies in LLMs.

- Systematic Evaluation Experiment:
The authors pose three clear research questions and provide a detailed experimental framework to evaluate whether separate alignment or common knowledge generalization methods could improve consistency between words and deeds.

**Weaknesses:**

- Ambiguity in the Underlying Hypothesis:
While the paper proposes that the inconsistency arises due to the separation of knowledge guiding words and deeds, this hypothesis remains underexplored. The authors could strengthen the paper by providing more in-depth analysis or empirical evidence that supports this claim. For example, conducting case studies to trace the model's reasoning paths, could illustrate where the divergence occurs.

**Questions:**

I have no questions for the authors.

---

> ### Author Response · Authors · 2024-11-25
> **Rebuttal**
>
> Dear Reviewer,
>
> We appreciate your recognition of our work, as this serves as great encouragement for us.
>
> Thank you for your invaluable suggestion. Considering that the primary goal of our study is to establish a benchmark for quantifying the inconsistency in LLMs between words and actions and to provide some widely used baselines to identify the universality and challenges experienced, we currently only provide a preliminary analysis of the causes of model inconsistency in sections 4.1 and 4.2. We agree with you that more in-depth analysis or empirical evidence that supports our hypothesis would indeed strengthen our paper. We plan to conduct a detailed exploration of this hypothesis in our future work, possibly including case studies as you suggested, to trace the model's reasoning paths and better illustrate where the divergence occurs.
>
> Best Regards.

---

> > ### Comment · Reviewer_eRDy · 2024-11-25
> >
> > Thank you for your thoughtful response and for outlining your plans for future work. I appreciate your efforts, and my score remains equally positive. Best regards.

---

### Official Review · Reviewer_C63h · 2024-11-07

**Soundness:** 4
**Presentation:** 2
**Contribution:** 4
**Rating:** 8
**Confidence:** 5

**Summary:**

This paper addresses the issue of inconsistencies in the texts generated by LLMs, formalized as the difference between the LLMs' "words" (i.e., replies to direct questions about the LLMs' views) and "deeds" (i.e., replies to questions about the preferred action in situations that manifest the LLMs' views). The authors create a new benchmark of matched word and deed questions and use it to evaluate a series of LLMs. They find evidence for rampant inconsistencies between the LLMs' words and deeds.

**Strengths:**

The problem that the paper addresses (i.e., inconsistencies between the LLMs' words and deeds) has been studied before, but a better understanding of the underlying issues is still lacking. In this context, the paper proposes a very valuable resource, both for future research on inconsistencies in LLMs and for evaluating the values encoded in LLMs. The analyses that the authors conduct using their benchmark are methodologically sound; the authors did a great job in testing the robustness of their findings.

**Weaknesses:**

The biggest weakness of the paper is that it completely ignores prior work on inconsistencies between the words and deeds of LLMs, claiming that it is "the first [study] to systematically evaluate the consistency of responses from prominent LLMs based on the words and deeds" (p. 10); while prior studies do not use the terms "words" and "deeds", they are still very similar in terms of motivation and overall findings, so I think it is critical that the authors add a comprehensive discussion and comparison before the paper can be published. There are two lines of work that I deem particularly relevant:

1) There has been a lot of work on the inconsistencies in the political values that LLMs express when asked in a multiple-choice setup ("words") versus more real-world use cases ("deeds"). For example, [Röttger et al. (2024)](https://aclanthology.org/2024.acl-long.816/) compare the behavior of LLMs when directly asked for their political values, and the values they implicitly support when asked to write texts about political issues, finding inconsistencies very similar to the ones described in this paper. There have been various other papers looking at the same question (e.g., [Moore et al., 2024](https://arxiv.org/abs/2407.02996)), none of which the authors cite.

2) The difference between the LLMs' "words" and "deeds" is related to the distinction between the implicit and explicit behavior of LLMs (e.g., [Bai et al., 2024](https://arxiv.org/abs/2402.04105), [Hofmann et al., 2024](https://www.nature.com/articles/s41586-024-07856-5)) -- another line of work that the authors should acknowledge. For example, Hofmann et al. (2024) show that the divergence between the explicit and the implicit behavior of LLMs is bigger for aligned than for unaligned models, which is exactly in line with the findings that the authors present (p. 7).

**Questions:**

Minor points:
- Table 1, caption: "a aligned -> "an aligned"
- Table 3: The abbreviation "IFT" is never introduced. Did you mean "SFT"?
- 421-422: "to generalize to generalize" -> "to generalize"
- Section 4.2, 2: I think Figure 5 does not clearly support your claim. For example, the bar for D2 is as high as the one for S1 with both models. You either need to provide a more detailed analysis or weaken your claim.
- Table 5: What is shown in the first column?

---

> ### Author Response · Authors · 2024-11-25
> **Rebuttal (part 1/3)**
>
> Dear Reviewer,
>
> Thank you for your recognition of our research value and for your detailed and constructive comments, which have been really valuable in enhancing the quality of the paper. We will respond to each of your questions in turn and we hope that our revisions and explanations will satisfy and address your previous concerns.
>
> > The biggest weakness of the paper is that it completely ignores prior work on inconsistencies ... that the authors present (p. 7).
>
> Thanks for your thoughtful reminder. We have conducted a comprehensive and thorough review of the relevant work, especially along the two directions you mentioned. **We have updated the discussion about the relevant work in section 6 of the new PDF.**
>
> **Survey Process**
>
> We adopted two survey methods:
> - We conducted a keyword-based search on Google Scholar. Specifically, we searched for these keywords: "LLM’s inconsistencies in values", "LLM’s inconsistencies in opinions", "LLM’s inconsistencies between what speaks and acts", "Do LLM make consistent choice across different answer settings", "LLM’s explicit and implicit behavior". We reviewed the first 80 articles under each keyword. With this method, we obtained 19 papers.
> - We complemented our search with a recursive search into the "related work" of the retrieved papers. In detail, we recursively reviewed the related work that were cited in the papers we found to address any potential oversights in our initial search. Using this method, we found an additional 5 relevant papers.
>
>
> **Comparison with Related Work**
>
> 1. Consistency of LLMs
>
> With LLMs demonstrating powerful capabilities in various tasks and gradually being deployed in real-world LLM applications, the consistency of LLM outputs has become a critical research direction. Generally, the consistency analysis falls into four categories: 1) Formal consistency, which analyzes the consistency of LLM outputs under different evaluation paradigms, such as multiple-choice questions and open-ended questions ``[1-4]``, different order of options in multiple-choice questions ``[5-7]``, etc.; 2) Semantic consistency, which measures the consistency of the model's responses under prompt variations with typical disturbances including paraphrases ``[4-5, 8-9]``, paragraph insertions ``[10]``, different languages ``[4, 11]`` and others ``[12-13]``; 3) Logical consistency, which measures models’ ability to make decisions without logical contradiction, including negational, symmetric, transitive, and additive consistency ``[14-15]``; 4) Factual consistency, measures models’ ability to generate outputs not contradictory to the common facts and given context ``[15]``. However, these studies mainly focus on the consistency of LLM's beliefs or facts in different application forms, but lacks analysis of the consistency of LLM's beliefs at different application depths. These two are different and even orthogonal research directions. To fill this gap, we propose a formal, multidomain consistency benchmark to quantitatively evaluate the model's inconsistency in words and deeds.
>
> 2. Implicit and explicit behavior of LLMs
>
> The distinction between the implicit and explicit behavior of LLMs has attracted much attention in navigating the ethics of AI, but most of them only focus on specific ethical issues, e.g., social bias and toxic language ``[16-24]``. Instead, the benchmark we propose investigates inconsistencies across multiple domains, including opinion versus action, non-ethical value versus action, ethical value versus action, and theory versus application. Of these, two have definite correct answers while the other two do not. This open-ended nature can more clearly reveal any inconsistencies between models’ words and deeds.
>
> **To conclude**, our study distinguishes itself through several key innovations:
> - The inconsistencies between words and deeds focus on models' consistency in a same belief across varying application depths, which differs from previous consistency studies.
> - The WDCT benchmark spans multiple domains, providing a comprehensive insight into LLMs’ inconsistencies of words and deeds.
> - We delve into the reasons behind these inconsistencies in LLMs, and conduct experiments to validate the role of alignment plays in amplifying the inconsistencies, which can offer valuable insights for future alignment endeavors.
>
> (You can find the references in the next part)

---

> ### Author Response · Authors · 2024-11-25
> **Rebuttal (part 2/3)**
>
> **Reference:**
>
>
> ``[1]`` Wang Y, Teng Y, Huang K, et al. Fake Alignment: Are LLMs Really Aligned Well? NAACL, 2024.
>
> ``[2]`` Röttger P, Hofmann V, Pyatkin V, et al. Political compass or spinning arrow? towards more meaningful evaluations for values and opinions in large language models.
>
> ``[3]`` Li W, Li L, Xiang T, et al. Can Multiple-choice Questions Really Be Useful in Detecting the Abilities of LLMs? COLING, 2024.
>
> ``[4]`` Moore J, Deshpande T, Yang D. Are Large Language Models Consistent over Value-laden Questions? EMNLP 2024.
>
> ``[5]`` Tjuatja L, Chen V, Wu T, et al. Do llms exhibit human-like response biases? a case study in survey design. TACL, 2024.
>
> ``[6]`` Pezeshkpour P, Hruschka E. Large Language Models Sensitivity to The Order of Options in Multiple-Choice Questions. NAACL, 2024.
>
> ``[7]`` Zheng C, Zhou H, Meng F, et al. Large language models are not robust multiple choice selectors. ICLR, 2023.
>
> ``[8]`` Bonagiri V K, Vennam S, Govil P, et al. SaGE: Evaluating Moral Consistency in Large Language Models. COLING, 2024.
>
> ``[9]`` Shu B, Zhang L, Choi M, et al. You don’t need a personality test to know these models are unreliable: Assessing the Reliability of Large Language Models on Psychometric Instruments. NAACL, 2024.
>
> ``[10]`` Kovač G, Sawayama M, Portelas R, et al. Large language models as superpositions of cultural perspectives.
>
> ``[11]`` Yao J, Yi X, Gong Y, et al. Value FULCRA: Mapping Large Language Models to the Multidimensional Spectrum of Basic Human Value. ACL: Human Language Technologies, 2024.
>
> ``[12]`` Rozen N, Bezalel L, Elidan G, et al. Do LLMs have Consistent Values?
>
> ``[13]`` Clymer J, Juang C, Field S. Poser: Unmasking Alignment Faking LLMs by Manipulating Their Internals.
>
> ``[14]`` Jang M, Lukasiewicz T. Consistency Analysis of ChatGPT. EMNLP, 2023.
>
> ``[15]`` Jang M, Kwon D S, Lukasiewicz T. BECEL: Benchmark for consistency evaluation of language models. ACL, 2022.
>
> ``[16]`` Hofmann V, Kalluri P R, Jurafsky D, et al. AI generates covertly racist decisions about people based on their dialect. Nature, 2024.
>
> ``[17]`` Bai X, Wang A, Sucholutsky I, et al. Measuring implicit bias in explicitly unbiased large language models.
>
> ``[18]`` Zhao Y, Wang B, Zhao D, et al. Mind vs. Mouth: On Measuring Re-judge Inconsistency of Social Bias in Large Language Models.
>
> ``[19]`` Wei Q, Chan A J, Goetz L, et al. Actions Speak Louder than Words: Superficial Fairness Alignment in LLMs. ICLR 2024 Workshop on Reliable and Responsible Foundation Models.
>
> ``[20]`` Dong X, Wang Y, Yu P, et al. Probing Explicit and Implicit Gender Bias through LLM Conditional Text Generation. Socially Responsible Language Modelling Research.
>
> ``[21]`` Giorgi S, Liu T, Aich A, et al. Modeling Human Subjectivity in LLMs Using Explicit and Implicit Human Factors in Personas. EMNLP Findings, 2024.
>
> ``[22]`` Xu C, Wang W, Li Y, et al. A study of implicit ranking unfairness in large language models. EMNLP Findings, 2024.
>
> ``[23]`` Wen J, Ke P, Sun H, et al. Unveiling the Implicit Toxicity in Large Language Models. EMNLP 2023.
>
> ``[24]`` Zhao Y, Wang B, Wang Y, et al. A Comparative Study of Explicit and Implicit Gender Biases in Large Language Models via Self-evaluation. COLING, 2024.

---

> ### Author Response · Authors · 2024-11-25
> **Rebuttal (part 3/3)**
>
> > Table 1, caption: "a aligned -> "an aligned"
>
> > 421-422: "to generalize to generalize" -> "to generalize"
>
> Thanks for your kind reminder. We have thoroughly reviewed and revised the writing of the paper, especially addressing the writing issues you pointed out.
>
> > Table 3: The abbreviation "IFT" is never introduced. Did you mean "SFT"?
>
> We apologize for the oversight. The abbreviation "IFT" stands for "Instruction Fine-tuning". We have now clarified this in the caption of Table 3.
>
> > Section 4.2, 2: I think Figure 5 does not clearly support your claim. For example, the bar for D2 is as high as the one for S1 with both models. You either need to provide a more detailed analysis or weaken your claim.
>
> Thanks for your thoughtful comment.
>
> We have conducted a more detailed analysis on the experimental results and infer that **the deviation might be due to a fewer number of newly aligned words in the later epochs, leading to instability in the consistency rate between words and deeds**.  Therefore, for more solid results, we have rerun the experiments three times. Based on the new experimental results, **we have redrawn the Figure, now named Figure 4 in section 4.2, and updated our conclusions accordingly:** the beliefs aligned during the initial stages of each alignment phase (SFT, DPO) are more likely to generalize to untargeted aspects more effectively.
>
> > Table 5: What is shown in the first column?
>
> We apologize for the confusion our previous descriptions might have caused.
>
> In the data augmentation experiment, we conducted alignment in three different data setups:
> - Base data: refers to the original data.
> - Non-augmented data: for this setup, we simply replicated the data four times to act as a baseline for comparison with the augmented data setup.
> - Augmented data: in this case, we paraphrased the data four times.
>
> **The first column of Table 5 displays the experimental results from the base data setup.** However, as it doesn't carry a strong comparability due to its distinct data volume to the other two setups, we have removed it in our new PDF.

---

> > ### Comment · Reviewer_C63h · 2024-11-26
> > **Reviewer Response**
> >
> > I thank the authors for their detailed rebuttal and especially the addition of the related work section. I have increased my score.

---

### Author Response · Authors · 2024-12-04
**Official Comment by Authors**

We thank all reviewers for the feedback!

**In this post**:

(1) We summarize positive things from the reviews.

(2) We summarize the changes to the updated PDF document.

**In the individual replies**, we address other comments.

### (1) Positive things

- **Significant research questions**
  - ```eRDy```: *The problem tackled is highly relevant ... as autonomous agents.*
  - ```8NCb```: *The authors address an important issue in evaluating words and deeds using LLMs.*

- **Valueable and novel benchmark**
  - ```C63h```: *the paper proposes a very valuable resource, both for future research on inconsistencies in LLMs and for evaluating the values encoded in LLMs.*
  - ```eRDy```: *The WDCT is a significant contribution, offering a well-structured approach to measure alignment between words and deeds across multiple domains (e.g., opinion, ethical values, theory). This benchmark provides a robust foundation for future alignment studies in LLMs.*
  - ```vDJr```: *This work proposes a novel benchmark called the WDCT to measure the inconsistency between words and actions.* and *The paper introduces two new metrics, ... offer valuable tools for evaluating model alignment.*
  - ```8NCb```: *The datasets introduced in the paper include several domain topics to ensure better generalisability.*

- **Extensive, detailed and sound analysis**
  - ```C63h```: *The analyses that the authors conduct using their benchmark are methodologically sound.*
  - ```eRDy```: *The authors pose three clear research questions and provide a detailed experimental framework to evaluate whether separate alignment or common knowledge generalization methods could improve consistency between words and deeds.*
  - ```8NCb```: *The habitual evaluation of LLMs is very extensive and detailed.*

- **Comprehensive robust testing**
  - ```C63h```: *the authors did a great job in testing the robustness of their findings.*

- **Presentation**
  - ```8NCb```: *The writing is clear, fluent and understandable.*

### (2) Major changes to the PDF

- Writing

  - Updated the introduction of the research question and its answer along with the contribution from different angles in section 1.

  - Updated the presentation of section 4, placing conclusions after research questions. The conclusions are organized into main conclusions (conclusion 1-3) and sub-conclusions (finding 1-6) for ease of comprehension.

  - A more comprehensive discussion on the related works in section 6.

  - Corrected the layout of tables and figures.

- Experiments

  - Results of more recent, advanced models, the LLaMA-3-8B and LLaMA-3-8B-Instruct in Table 3.

  - A data augmentation method by automatically generating aligned data in section 4.3.2.

  - An analysis of the correlation between the data size and evaluation results in Appendix B.1.

  - An analysis of the impact of LLM constructors.

  - An analysis of the sensitivity of the consistency scores to the prompts.

  - An analysis of the root causes of word-action inconsistency.

Once the page limit is confirmed, we will include the further experiments in the final paper.

---

### Meta-Review · Area_Chair_QyJg · 2024-12-19

**Metareview:**

The paper proposes a new benchmark Words and Deeds Consistency Test (WDCT) to examine LLM consistency between word-based (direct query about opinion) and deed-based (response to a situation) scenarios. Reviewers generally agree that the benchmark is timely and exposes a interesting model behavior. Extensive analysis are conducted in the paper to stud this phenomenon. The authors should add more explicit and clearer discussion about how this benchmarks tests a notion of consistency that is fundamentally different from those in prior work (e.g. consistency under prompt variance, etc.)

**Additional Comments On Reviewer Discussion:**

Reviewer 8NCb raised several issues about the writing and presentation of this paper. In my view, most of these were resolved. Other reviewers (C63h) raised issues about better contextualizing this study with prioir works in similar spaces, which the authors address in the rebuttal.

---

### Decision · Program_Chairs · 2025-01-22

Accept (Poster)